# Structure Modulations and Symmetry of Lazurite-Related Sodalite-Group Minerals

**Nadezhda B. Bolotina** [1,2,*] , **Anatoly N. Sapozhnikov** [3] , **Nikita V. Chukanov** [2,4] and **Marina F. Vigasina** [2]

1  Shubnikov Institute of Crystallography, Federal Scientific Research Centre "Crystallography and Photonics", Russian Academy of Sciences, Leninsky Avenue 59, 119333 Moscow, Russia

2  Faculty of Geology, Moscow State University, Vorobievy Gory, 119991 Moscow, Russia

3  Vinogradov Institute of Geochemistry, Siberian Branch of Russian Academy of Sciences, Favorskii Street 1a, 664033 Irkutsk, Russia

4  Federal Research Center of Problems of Chemical Physics and Medicinal Chemistry, Russian Academy of Sciences, 142432 Chernogolovka, Russia

*  Correspondence: bolotina@ns.crys.ras.ru or nb_bolotina@mail.ru

**Abstract:** Lazurite and other lazurite-related minerals (LRMs) containing sulfur in both sulfate and sulfide forms are sodalite-type compounds with various extraframework species, of which the tendency to order leads to structural modulations with a period that is either commensurate or incommensurate with the period of the basic lattice. In this work, the structures of incommensurately modulated monoclinic LRMs are re-examined based on the superstructure of slyudyankaite, formerly known as triclinic lazurite. Similarities and differences between three one-dimensionally modulated LRMs and cubic LRM structures modulated in several directions are discussed. Assumptions are made on how the symmetry of the structure and the composition of the crystal can affect the period of structural modulation.

**Keywords:** modulated crystals; sodalites; lazurite-related minerals; X-ray diffraction; spectroscopy





## 1. Introduction

The structures of natural and synthetic aluminosilicates belonging to the topological type of sodalite are based on a close packing of 24-vertex polyhedra in the form of truncated octahedra that consist of Al and Si atoms (Figure 1a). In most cases, Al and Si occur in almost equal amounts, and their positions alternate, i.e., each Al atom is surrounded by Si atoms and vice versa. Each Al and Si atom at the vertex of a truncated octahedron is in a tetrahedral environment of oxygen atoms. A fragment of a three-dimensional framework formed by vertex-connected tetrahedra $TO_4$ ($T$ = Al, Si) is shown in Figure 1b. Eight six-membered and six four-membered rings of tetrahedra frame a cavity characteristic of sodalite-type compounds (a so-called sodalite cage).

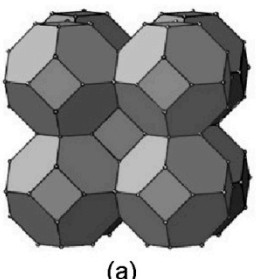
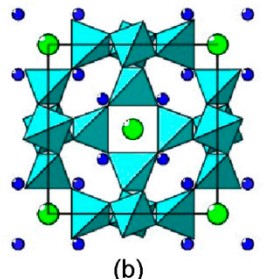

**Figure 1.** (**a**) Sodalite-type structure, shown as a close packing of truncated octahedra. Dots denote centers of the $TO_4$ tetrahedra. (**b**) Structure of sodalite, $Na_8[Al_6Si_6O_{24}]Cl_2$, in polyhedral presentation of the framework (projection onto the face of a cubic unit cell). Large green circles are $Cl^-$ anions and small blue circles are $Na^+$ cations.

The ancestor of this structural type is sodalite *s.s.*, $Na_8[Al_6Si_6O_{24}]Cl_2$. This mineral crystallizes in a cubic system with a unit-cell parameter $a_{cub} \approx 9$ Å, Z = 1 [1]. Each cavity in the sodalite framework contains a $Cl^-$ anion in a tetrahedral environment of four $Na^+$ cations which are displaced into the cavity and limited by the planes of the sixfold rings.

The chemical compositions of extraframework components (cations, anions, radical anions, and neutral molecules) in other minerals of the sodalite group are much more diverse. The tendency of extraframework species to order often leads to structure modulations with a period that is either commensurate or incommensurate with the period of the basic crystal lattice. From the end of the 19th century and up until recently, the name "lazurite" was assigned to minerals of the sodalite group containing significant amounts of sulfide sulfur. In 1985, based on the results of a structural analysis of two samples, the composition of lazurite was characterized by the simplified formula $Na_6Ca_2(Al_6Si_6O_{24})S_2$ [2], although, according to the same authors, the studied samples contained sulfate ions, the number of which (in moles) exceeded the number of $S^{2-}$ ions. A different formula for lazurite, $(Na,Ca)_{7-8}(Si_6Al_6O_{24})(SO_4,S,Cl)_2 \cdot H_2O$ is given in the reference book [3]. Recently, the Commission on New Minerals, Nomenclature and Classification of the International Mineralogical Association (IMA CNMNC) left the name "lazurite" to refer to a cubic mineral whose composition corresponds to the simplified formula $Na_7Ca(Al_6Si_6O_{24})(SO_4)S_3^{\bullet-} \cdot nH_2O$, where "$\bullet$" denotes an unpaired electron [4]. The presence of a large amount of $S_3^{\bullet-}$ is the cause of the dark blue color of lazurite. The other members of the sodalite group that contain sulfur in both sulfate and sulfide forms, which were formerly classified as lazurites, are now known as lazurite-related minerals (LRMs). All of them, similar to lazurite, are characterized by a variety of structural modulations. Various species, including $Na^+$, $K^+$, $Ca^{2+}$, $H_3O^+$, $CO_2$, COS, $S_4$, $S_6$, $S_2^{\bullet-}$, $S_3^{\bullet-}$, $SO_4^{2-}$, $CO_3^{2-}$, $SO_3^{2-}$, $S^{2-}$, $Cl^-$, and $F^-$, have been detected in LRMs by using a complex of spectroscopic methods [4–10]. The presence of significant amounts of sulfide sulfur is typical for LRMs from gem lazurite deposits. A visual representation of the relationship between the content of $S_3^{\bullet-}$ in the composition and the intensity of the blue color of LRMs is provided by illustrations in [6].

Optically isotropic LRMs have cubic structures that are modulated in several directions. The X-ray diffraction patterns of these crystals are significantly complicated by satellites that are oriented in several directions from the main reflections. The average structure of cubic lazurite (according to the old classification) has been determined in [2,11], using only the main reflections. The modulated structure of cubic lazurite was later considered in the (3+2)D and (3+3)D spaces [12–14], which made it possible to ascertain an idea of the framework modulations, but the behavior of atoms in the cavities of the framework was difficult to model.

Optically anisotropic LRMs with noncubic, one-dimensionally modulated structures are much less common. The main reflections form a pseudocubic reciprocal lattice with the averaged period $a^*_{cub}$ (square bounded by black lines in Figure 2). X-ray diffraction patterns from these crystals are more conveniently indexed in an orthorhombic or pseudo-orthorhombic setting: $a^* \approx a^*_{cub}$ and $b^* \approx c^* \approx a^*_{cub}\sqrt{2}/2$ (square bounded by red lines in Figure 2). The periods of the corresponding direct lattice are $a \approx a_{cub}$ and $b \approx c \approx a_{cub}\sqrt{2}$.

Superstructural reflections (blue circles in Figure 2) indicate a commensurate modulation of the structure of triclinic LRMs with the wave vector $\mathbf{q} = 0.5\mathbf{c}^*$ (Figure 2a) and orthorhombic LRMs with the wave vector $\mathbf{q} = 0.33\mathbf{c}^*$ (Figure 2c). The commensurately modulated structures of triclinic [15] and orthorhombic [16] LRMs were studied in 3D as superstructures, with the periods of $2c$ and $3c$, respectively. The unit cells of the corresponding reciprocal lattices are highlighted in gray in Figure 2a,c. Later, the composition, structure, and properties of triclinic and orthorhombic LRMs were reinvestigated in [10] and [8,17], respectively. Specific features that made it possible to approve both crystals as new mineral species were revealed. Triclinic LRMs were called slyudyankaite [10] and orthorhombic LRMs received the name vladimirivanovite [8].

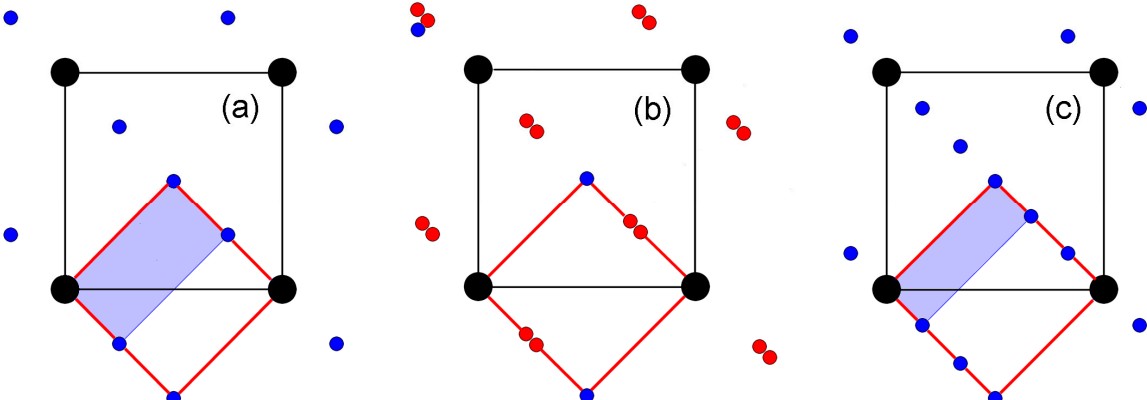

**Figure 2.** Model fragments of diffraction patterns from triclinic (**a**), monoclinic (**b**), and orthorhombic (**c**) LRMs in projection along the $a^*$ axis of the reciprocal lattice. Large black circles are the main reflections at the sites of the pseudocubic reciprocal lattice, blue circles are superstructural reflections, and red circles are satellites at distances of $\approx 0.43 c^*$ from both the main and superstructural reflections.

The structure modulation of monoclinic LRMs can be equally justified as being incommensurate, with the vector $\mathbf{q} \approx 0.43\mathbf{c^*}$, or commensurate, with the vector $\mathbf{q} = (3/7)\mathbf{c^*}$, $3/7 \approx 0.4286$. Even if the modulation of its structure were commensurate, the transition to a superstructural cell with a period of $7c$ would hardly be convenient for structural analysis. Because of this, its structure was initially considered in the basic unit cell with the parameters $a \approx a_{cub}$, $b \approx c \approx a_{cub}\sqrt{2}$, using the (3+1)D model. The average structure of monoclinic LRMs was studied earlier [18] and the first data on framework modulation were obtained [19]; however, complex modulations of the extraframework cations $Na^+$ and $Ca^{2+}$, as well as the behavior of sulfate and sulfide sulfur, have remained poorly understood. The structure of slyudyankaite, which has recently been studied in great detail [10], served as an incentive to resume studies of LRM structures. The diffraction patterns from monoclinic LRMs (Figure 2b) are easily modeled by splitting the superstructural reflections from slyudyankaite (Figure 2a) on the one-quarter and three-quarter diagonals. The period of the modulation wave is $2c$ for the triclinic LRM (slyudyankaite), $\approx 2.33c$ for the monoclinic LRM, and $3c$ for the orthorhombic LRM (vladimirivanovite). These three structurally related crystals are also similar in chemical composition (see below). There is most likely much in common in the characterization of the modulations, not only of framework atoms but also of extraframework components.

In this work, the structure of the monoclinic LRM is re-examined based on the structure of slyudyankaite. Similarities and differences between three one-dimensionally modulated structures and cubic LRM structures modulated in several directions are discussed. It is suggested that the symmetry of the structure and the composition of the crystal can affect the period of the structural modulation.

## 2. Samples and Experimental Methods

All samples of LRMs investigated and discussed in this work originated from gem lazurite deposits in the Baikal Lake area, Siberia, Russia. Along with optically isotropic cubic sodalite-group minerals, their orthorhombic, monoclinic, and triclinic analogues were identified in these deposits. These anisotropic LRMs occur in zones of high-temperature recrystallization of common lazurite-bearing rocks, in association with cubic LRMs, calcite, diopside, pyrite, fluorapatite, and, occasionally, phlogopite. Their short descriptions, including chemical formulas, are given below. For uniformity, all formulas were reduced to the volume of a pseudocubic cell and supplemented by the number of formula units Z per unit cell volume of a given crystal. Extraframework components were identified using electron probe microanalysis, various spectroscopic methods, and X-ray structure analysis. More detailed data can be found in the cited publications.

Sample 1 is the holotype of lazurite with the empirical formula $(Na_{6.97}Ca_{0.88}K_{0.10})$ $(Si_{6.04}Al_{5.96}O_{24})(SO_4)_{1.09}(S_3^{\bullet-})_{0.55}S^{2-}_{0.05}Cl_{0.04}\cdot 0.72H_2O$, $Z = 1$ [4]. The mineral is characterized by commensurate and incommensurate modulations. The *a* parameter of the cubic cell is equal to 9.087(3) Å. Sample 1 is slightly optically anisotropic, with $\alpha' = 1.523(2)$ and $\gamma' = 1.525(2)$. The color is dark blue due to the presence of a large amount of $S_3^{\bullet-}$ radical anions.

Sample 2 is the holotype of the orthorhombic LRM with the simplified formula $Na_6Ca_2(Si_6Al_6O_{24})(SO_4,S_3^{\bullet-},S_2^{2-},Cl)\cdot H_2O$, $Z = 6$. It was approved by IMA CNMNC as a new mineral with the name vladimirivanovite [8]. Sample 2 is optically anisotropic and biaxial, with variable refractive indices. The color is deep blue.

Sample 3 is a monoclinic LRM. Its chemical composition, determined by means of electron microprobe, wet chemical analyses, and IR and Raman spectroscopy (see below), and taking into account the charge balance requirement, is as follows (wt.%): $Na_2O$ 17.77, $K_2O$ 0.88, $CaO$ 7.40, $Al_2O_3$ 27.35, $Fe_2O_3$ 0.22, $SiO_2$ 32.76, $SO_3$ 10.89, $S^{2-}$ 0.42, $S_3^{\bullet-}$ 1.52, $Cl$ 0.19, $CO^2$ 0.20, $-O=(Cl^-,S^{2-},S_3^{\bullet-})$ $-0.38$, total 99.22. The empirical formula is $Na_{6.35}Ca_{1.46}K_{0.21}(Si_{6.04}Al_{5.94}Fe_{0.02}O_{24})(SO_4^{2-})_{1.505}(S^{2-})_{0.14}(S_3^{\bullet-})_{0.17}Cl_{0.06}(CO_2)_{0.05}\cdot nH_2O$ with $n < 1$, $Z = 2$. The color is blue.

Sample 4 is the holotype of the triclinic LRM with the simplified formula $Na_7Ca_1$ $(Si_6Al_6O_{24})(SO_4)_{1.5}(S_6)_{1/12}(CO_2)_{0.25}\cdot 0.5H_2O$, $Z = 4$. It was approved by the IMA CNMNC as a new mineral with the name slyudyankaite [10]. According to Raman spectroscopy data (see below), slyudyankaite contains minor admixtures of $S_4$ molecules and $S_4^{\bullet-}$ radical anions, as well as trace amounts of $HS^-$ anions. Sample 4 is optically anisotropic and biaxial. The refractive indices are $\alpha = 1.506$, $\beta = 1.509$, and $\gamma = 1.513$. The color varies from green to light blue and pink because of the presence of $S_6$ (a weak yellow chromophore), as well as trace amounts of $S_3^{\bullet-}$ and $S_4$ (strong blue and red chromophores, respectively).

Sample 5 is a light blue cubic LRM (haüyne) with an *a* parameter of 9.077 Å and the empirical formula $(Na_{6.45}Ca_{1.35}K_{0.03})(Al_{5.93}Si_{6.07}O_{24})(SO_4)_{1.35}(SO_3)_{0.37}(S_2)_{0.02}Cl_{0.16}\cdot nH_2O$, $Z = 1$ [6].

In order to obtain IR absorption spectra, powdered samples were mixed with anhydrous KBr, pelletized, and analyzed using an ALPHA FTIR spectrometer (Bruker Optics) at a resolution of 4 cm$^{-1}$. Sixteen scans were collected for each spectrum. The IR spectrum of an analogous pellet of pure KBr was used as a reference.

Raman spectra were obtained for randomly oriented grains using an EnSpectr R532 spectrometer based on an OLYMPUS CX 41 microscope coupled with a diode laser ($\lambda = 532$ nm) at room temperature (Moscow State University, Faculty of Geology). The spectra were recorded in the range from 100 to 4000 cm$^{-1}$ with a diffraction grating (1800 gr mm$^{-1}$) and a spectral resolution of about 6 cm$^{-1}$. The output power of the laser beam was in the range from 5 to 13 mW. The diameter of the focal spot on the sample was 5–10 μm. The backscattered Raman signal was collected with a $40^\times$ objective; the signal acquisition time for a single scan of the spectral range was 1 s, and the signal was averaged over fifty scans. Crystalline silicon was used as a standard.

Chemical analyses of Sample 3 were carried out using a JXA_8200 Jeol electron microprobe equipped with a high-resolution scanning electron microscope, an energy dispersion system (EDS), a SiLi detector with a resolution of 133 eV, and a wave dispersion spectrometer (WDS). The chemical composition was measured with the WDS operated at an acceleration voltage of 20 kV, with a current intensity of 10 nA and a counting time of 10 s. The beam was defocused to 20 μm to decrease the thermal effect on the sample. Under these conditions, the mineral was stable with respect to the beam effect. The following standards and analytical lines were used: pyrope (Si, Kα), albite (Al, Na, Kα), diopside (Ca, Kα), orthoclase (K, Kα), barite (S, Kα), and Cl-apatite (Cl, Kα). The contents of the elements were calculated using the ZAF procedure. Sulfate sulfur was determined via conventional wet chemical analysis using acidic decomposition. Sulfide sulfur was accepted as the difference between the total sulfur and the sulfate sulfur. The content of $CO_2$ was determined from the IR spectrum using a method described in [5].

Single-crystal X-ray diffraction data for monoclinic LRMs (sample 3) were collected on an Xcalibur diffractometer (Oxford Diffraction), but they showed a rather low quality, which was insufficient for a detailed analysis of the structure. Visualization of the X-ray diffraction pattern using the Evald procedure (CrysAlisPro, version 171.38.43) made it possible to detect only shifts in a part of the reflections from the lattice sites, which was a sign of structural modulation with a wave vector $q < 0.5c^*$ in a pseudo-orthorhombic setting. In experiments carried out in 2006–2007, another sample of a similar chemical composition ($Na_{6.63} Ca_{1.26} K_{0.04} (Al_6Si_6O_{24})(SO_4)_{1.53} (S_3)_{0.33} Cl_{0.05}$) and originating from the same deposit was used. Diffraction data for structural analysis were then obtained on a CAD-4 diffractometer (Enraf Nonius) with a point detector. The parameters of the basic monoclinic cell were $a = 9.0692(1)$ Å, $b = 12.8682(1)$ Å, $c = 12.8725(1)$ Å, and $\gamma = 90.186(1)°$; the modulation wave vector was $q \sim 0.43c^*$. To achieve better data resolution in the diffraction pattern, intensities of 20,385 reflections, including 6778 main and 13,607 first-order satellites, were measured using CuKα radiation (λ = 1.54178 Å). Main reflections and first-order satellites were brought, one after another, on an Ewald sphere. The integral intensities were measured in the angular interval ($\omega - \Delta\omega$, $\omega + \Delta\omega$) using a special technique. Before being measured, each reflection placed on the Ewald sphere was rotated through a precalculated angle ψ around the scattering vector **H** to prevent the superposition of the intensities of close reflections (Figure 2b). Taking absorption into account, the shape of the sample was approximated using a sphere 0.1 mm in diameter.

Fragments of the measured diffraction pattern in the planes $h = 0$ and $h = 4$ of the reciprocal lattice were modeled using the Jana program [20] according to the experimental data (Figure 3), which were used in this work to refine the structure model of monoclinic LRMs. The previous model [19] remained incomplete due to the ambiguous interpretation of the modulation of extraframework atoms, but now this problem has been solved.

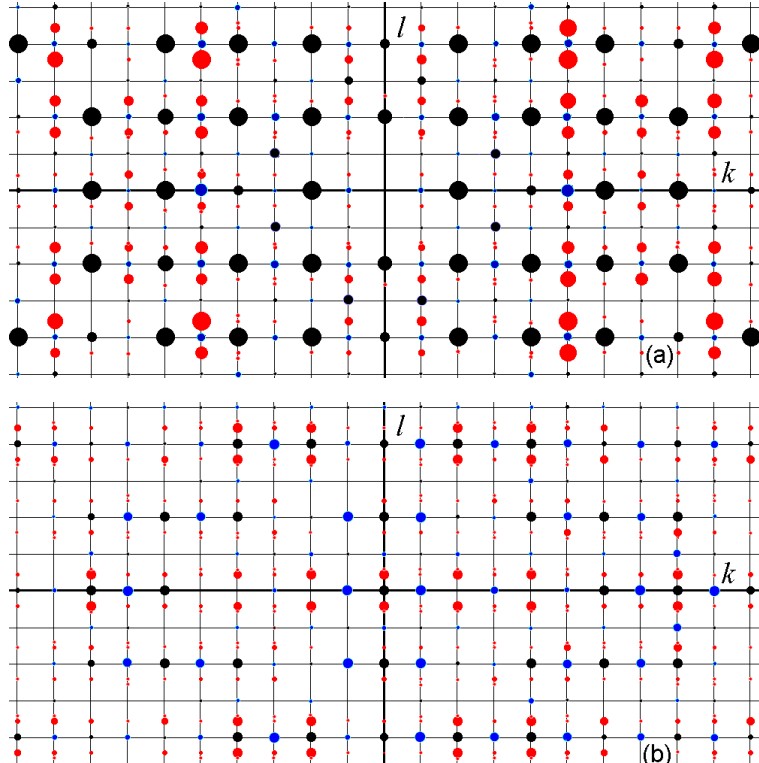

**Figure 3.** Modeled fragments of the diffraction pattern from monoclinic LRMs in the planes $h = 0$ (**a**) and $h = 4$ (**b**) of the reciprocal lattice. The main, superstructural, and satellite reflections are indicated by black, blue, and red circles, respectively. The size of the circle is proportional to the intensity of the reflex.

### 3. Vibrational Spectroscopy

Vibrational (infrared and Raman) spectra of Samples 1 through 4 are presented in Figures 4 and 5. Positions of bands and their assignments are given in Tables 1 and 2. The assignment of the bands was made based on data from [5,6,9,10,21–27].

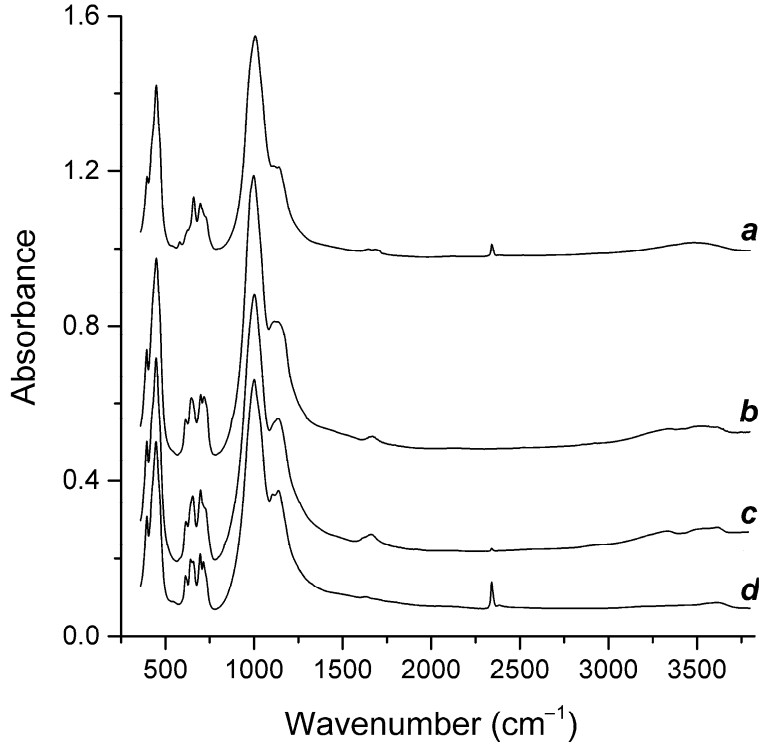

**Figure 4.** IR spectra of Samples 1 through 4 (the curves a through d, respectively).

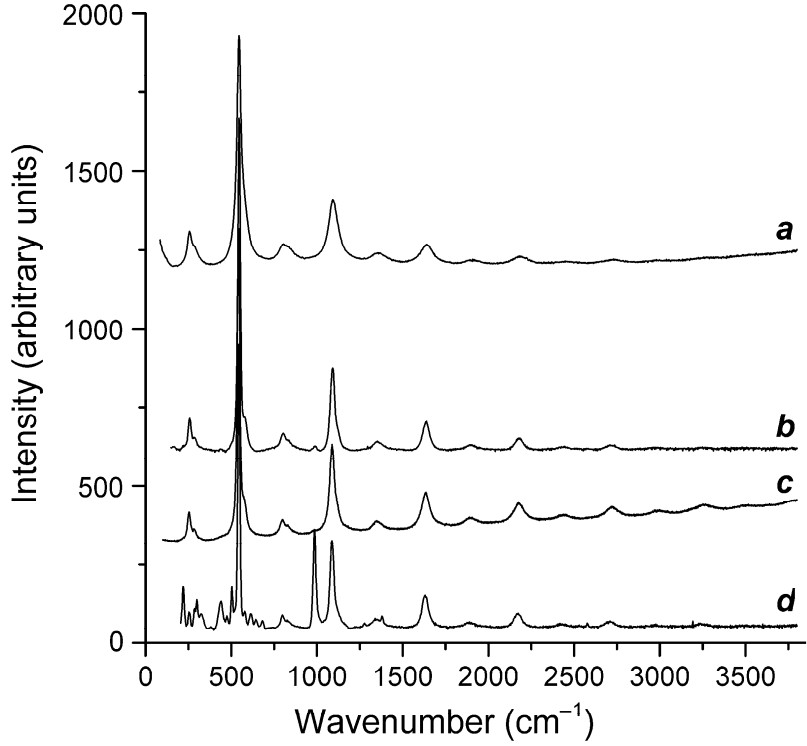

**Figure 5.** Raman spectra of Samples 1 through 4 (the curves a through d, respectively).

**Table 1.** Wavenumbers of IR absorption bands ($cm^{-1}$) and their assignments.

| Sample 1 | Sample 2 | Sample 3 | Sample 4 | Assignment |
|---|---|---|---|---|
| 448 s, 395 | 449 s, 396 | 448 s, 395 | 447 s, 396 | Lattice modes involving framework bending and liberations of extraframework groups |
| | | | 542 w | $S_6$ stretching mode |
| 580 w | | | | Antisymmetric stretching vibrations of the $S_3^{\bullet-}$ radical anions |
| 619 | 614 | 617 | 614 | Bending vibrations of the $SO_4^{2-}$ anionic groups (the $F_2(\nu_4)$ mode) |
| | | | 642 | Stretching vibrations of the neutral $S_4$ molecule having cis conformation |
| 709, 697, 655 | 718, 669, 647 | 725 sh, 699, 654 | 715, 697, 656 | Mixed vibrations of the aluminosilicate framework |
| 1000 s | 999 s | 1003 s | 1002 s | Stretching vibrations of the aluminosilicate framework |
| 1138, 1095 | 1133, 1116 | 1137 | 1138, 1107 | Asymmetric stretching vibrations of the $SO_4^{2-}$ anionic groups (the $F_2(\nu_3)$ mode) |
| 1683 w, 1622 w | 1665 w | 1660, 1620 sh | 1632 w | Bending vibrations of the $H_2O$ molecules |
| | | | 2040 w | Stretching vibrations of COS molecules |
| 2342 w | | 2342 w | 2385 w *, 2341 | Antisymmetric stretching vibrations of $CO_2$ molecules |
| 3415 | 3605 sh, 3545, 3343 | 3617, 3530 sh, 3330 | 3610 w | O–H stretching vibrations of $H_2O$ molecules |

Note: w—weak band, s—strong band, sh—shoulder. * The band at 2385 $cm^{-1}$ corresponds to antisymmetric stretching vibrations of $^{13}CO_2$.

**Table 2.** Wavenumbers of Raman bands ($cm^{-1}$) and their assignment.

| Sample 1 | Sample 2 | Sample 3 | Sample 4 | Assignment |
|---|---|---|---|---|
| | | | 219 | Combination of low-frequency lattice modes |
| 257 | 258 | 253 | 255 | $S_3^{\bullet-}$ bending mode ($\nu_2$) |
| 285 w | 284 w | 283 w | 284 | Combination of low-frequency lattice modes involving $Na^+$ cations and/or $S_6$ bending mode |
| | | | 299 | $S_4^{\bullet-}$ bending vibrations |
| | | | 320 | *cis*-$S_4$ mixed $\nu_4$ mode (combined symmetric bending + stretching vibrations) |
| | | | 380 w | *cis*-$S_4$ mixed $\nu_3$ mode |
| | 438 w | | 437 | $SO_4^{2-}$ (the $E(\nu_2)$ mode) and/or $\delta$(O–Si(Al)–O) bending vibrations |
| | | | 477 w | $S_6$ stretching mode and/or mixed $\nu_4$ mode of *trans*–$S_4$ or $S_4^{2-}$ |
| | | | 503 | Bending vibrations of four-membered aluminosilicate rings belonging to the framework |
| 546 s | 545 s | 542 s | 545 s | $S_3^{\bullet-}$ symmetric stretching ($\nu_1$) mode |
| 585 sh | 576 sh | 570 sh | 580 w | $S_3^{\bullet-}$ antisymmetric stretching ($\nu_3$), possibly, overlapping with the stretching band of $S_2^{\bullet-}$ |
| | | | 614 w | $SO_4^{2-}$ bending vibrations ($F_2(\nu_4)$ mode) and/or $S_2^{\bullet-}$ stretching mode |
| | | | 646 w | *cis*-$S_4$ symmetric stretching mode |

**Table 2.** *Cont.*

| Sample 1 | Sample 2 | Sample 3 | Sample 4 | Assignment |
|---|---|---|---|---|
| | | | 682 w | *trans*-$S_4$ symmetric stretching $\nu_3$ mode |
| 811 | 804 | 799 | 808 | $S_3^{\bullet-}$ combination mode ($\nu_1 + \nu_2$) |
| | 825 sh | 826 w | | Framework stretching vibrations? |
| | 989 w | 986 w | 985 s | $SO_4^{2-}$ symmetric stretching vibrations ($A_1(\nu_1)$ mode) |
| 1093s | 1092s | 1088s | 1088 s | $S_3^{\bullet-}$ overtone ($2'\nu_1$) |
| | | | 1279 w | $CO_2$ Fermi resonance |
| | | | 1340 | Symmetric C–O stretching vibrations of $CO_2$ molecules involved in strong dipole–dipole interactions with $H_2O$ molecules |
| 1363 | 1352 | 1348 | | $S_3^{\bullet-}$ combination mode ($2\nu_1 + \nu_2$) |
| | | | 1381 | $CO_2$ Fermi resonance |
| 1638 | 1639 | 1636 | 1631 | $S_3^{\bullet-}$ overtone ($3\times\nu_1$) |
| 1903 | 1903 w | 1891 | 1891 w | $S_3^{\bullet-}$ combination mode ($3\times\nu_2 + \nu_1$) |
| 2181 | 2178 | 2175 | 2172 | $S_3^{\bullet-}$ overtone ($4\times\nu_1$) |
| 2440 w | 2435 w | 2438 w | 2428 w | $S_3^{\bullet-}$ combination mode ($4\times\nu_2 + \nu_1$) |
| | | | 2575 w | $HS^-$ stretching mode |
| 2720 | 2709 | 2721 | 2710 | $S_3^{\bullet-}$ overtone ($5\times\nu_1$) |
| | 2975 w | 2985 w | 2964 w | $S_3^{\bullet-}$ combination mode ($5\times\nu_1 + \nu_2$) |
| 3260 w | 3240 w | 3252 | 3247 w | $S_3^{\bullet-}$ overtone ($6\times\nu_1$) |
| | | 3490 w | | $H_2O$ stretching vibrations |

The IR spectra are dominated by strong bands of the aluminosilicate framework, and the Raman spectra are dominated by strong bands of the $S_3^{\bullet-}$ radical anion. However, other (mainly weak) bands bear important information on extraframework anions and neutral molecules.

According to the IR spectroscopy data, all studied samples contained $H_2O$ molecules and $SO_4^{2-}$ anionic groups. In Samples 1, 2, and 3, $S_3^{\bullet-}$ was the main polysulfide species. In Sample 4, extraframework anions and neutral molecules were much more diverse and included significant amounts of $S_2^{\bullet-}$, $S_3^{\bullet-}$, $S_4^{\bullet-}$, $HS^-$, $S_4$, $S_6$, COS, $CO_2$, and $H_2O$.

## 4. Structure Model and Modulations of Monoclinic LRMs

### 4.1. Refinement of the Structure Model

After averaging the measured intensities in the Laue class $2/m$, the structure of a monoclinic LRM with incommensurate modulation was studied in the (3+1)D symmetry group $P11a(00\delta)0$, $\delta \approx 0.43$, using main $hkl0$ reflections and first-order $hklm$ satellites ($m = \pm1$). Thermal atomic vibrations were refined in the isotropic approximation for all atoms in order to eliminate the correlation between the anisotropy of thermal vibrations and the amplitudes of strong positional modulations. The main information about the crystal, the experiment, and the results of the refinement of the structural model is presented in Table 3.

**Table 3.** Data on the single-crystal XRD experiment and structure refinement details.

| Crystal Data | |
|---|---|
| Chemical formula | $Na_{6.63} Ca_{1.26} K_{0.04} (Al_6Si_6O_{24})(SO_4)_{1.53} (S_3)_{0.33} Cl_{0.05}$ |
| Z | 2 |
| Crystal system, (3+1)D space group | monoclinic, $P11a(00\delta)0$ * |
| Modulation wavevector | $\mathbf{q} \approx 0.43\mathbf{c}^*$ |
| Temperature (K) | 293 |
| $a, b, c$ (Å); $\gamma$ (deg) | 9.0692(1), 12.8682(1), 12.8725(1); 90.186(1) |
| V (Å$^3$) | 1502.27(1) |
| Radiation type | CuK$\alpha$, $\lambda$ = 1.54178 Å |
| Data collection | |
| $R_{int}$ (obs/all) (%) | 8.02/8.95 for 5386/9139 reflections averaged from 11,590/20,385 reflections |
| $(\sin \theta/\lambda)_{max}$ (Å$^{-1}$) | 0.626 |
| Refinement | Based on $F$ |
| $R_{obs}$ [$F > 3\sigma(F)$], $wR_{obs}$ ($F$) (%) | 7.62, 7.71 |
| $R_{obs}$ ($F$), $wR_{obs}$ ($F$) (%) for main reflections | 6.74, 7.19 |
| $R_{obs}$ (F), $wR_{obs}$ ($F$) (%) for satellites | 8.42, 8.11 |
| Weighting_scheme | $w = 1/[\sigma^2(F)+(0.01F)^2]$ |
| No. of parameters | 462 |
| No. of restraints | 30 |
| No. of constraints | 72 |
| $\Delta\rho_{max}$, $\Delta\rho_{min}$ (e Å$^{-3}$) | 1.04, −0.80 |

* Symmetry operators: (1) x1, x2, x3, x4; (2) x1 + 1/2, x2, −x3, −x4.

### 4.2. Positional Modulations in the Structure of Monoclinic LRMs

Positional (displacive) modulations for most atoms of a monoclinic LRM are defined in the structural model using harmonic wave functions. Waves with the vector $\mathbf{q} = \delta\mathbf{c}^*$ determine in 3D the displacements ($dx$, $dy$, $dz$) of an atom from the lattice site $\mathbf{r}_0(x_0, y_0, z_0)$ to $\mathbf{r}(x = x_0 + dx, y = y_0 + dy, z = z_0 + dz)$, which is $\mathbf{r}(x1 = x, x2 = y, x3 = z, x4 = \mathbf{q}\cdot\mathbf{r}_0)$ in the space (3+1)D.

Graphs in Figures 6 and 7 show the displacements ($dx$, $dy$, $dz$) of Al, Si, and O framework atoms from the lattice sites along each of the three coordinates ($x$, $y$, $z$) as a function of the fourth coordinate (modulation wave phase) for a period of $0 \leq x4 \leq 1$. As can be seen from the figures, all atoms in the framework undergo strong positional modulations, i.e., they are well-matched in phase periodic displacements from the sites of the basic lattice. The displacements of the Al and Si atoms along the $x1$ coordinate are especially well phase-matched (Figure 6a). Other displacements of the Al and Si atoms are not so large in amplitude, but they coincide at the x3 coordinate (Figure 6c). As expected, the positional modulations of oxygen atoms (Figure 7) at the vertices of the $TO_4$ tetrahedra are phase-matched and are especially strong along the $x1$ coordinate, reaching displacements of $\pm0.8$ Å in amplitude for some atoms (Figure 7a).

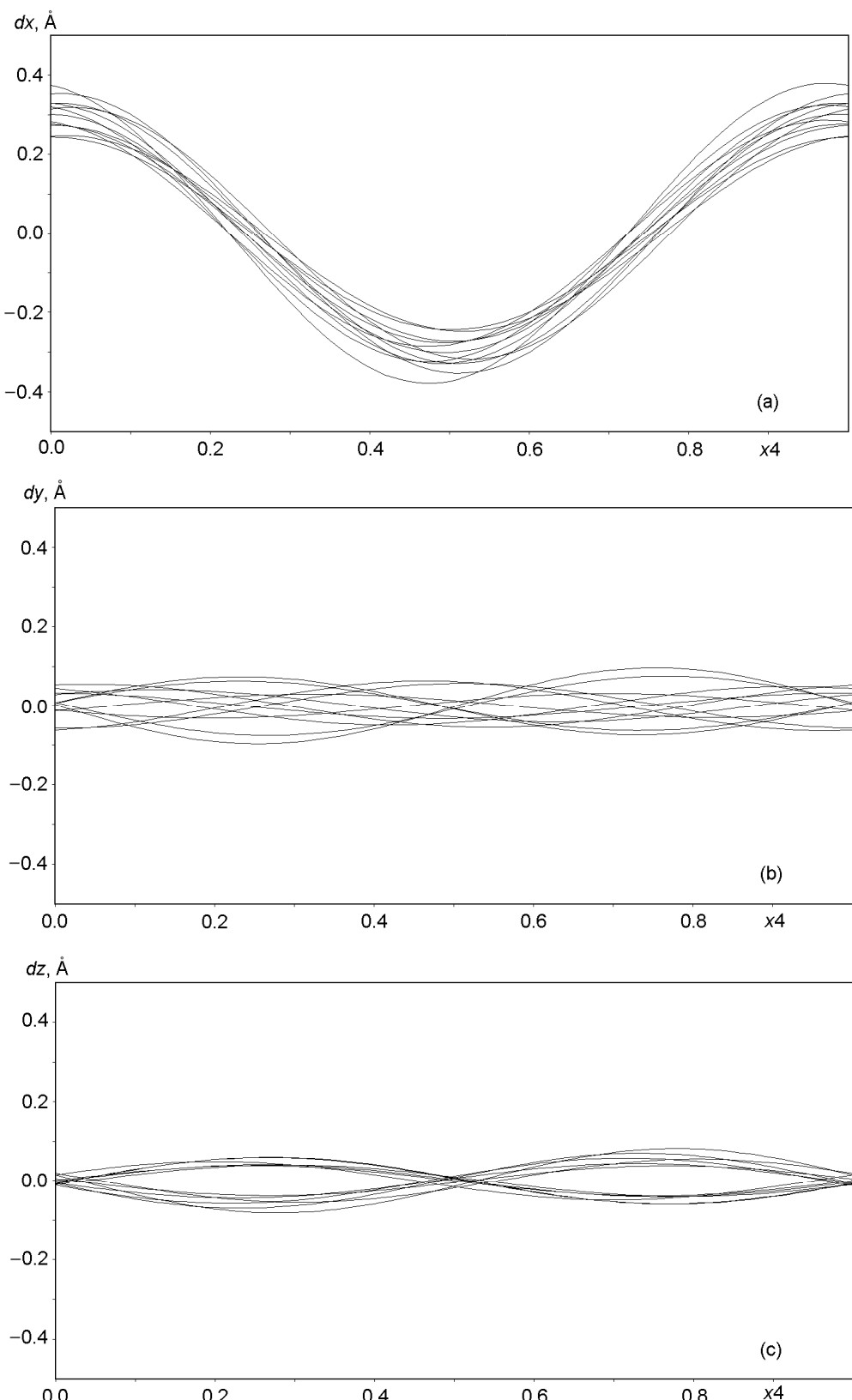

**Figure 6.** Displacements of Al and Si atoms from sites in the basic lattice on a modulation wave period of $0 \leq x4 \leq 1$: (**a**) *dx* along $x = x1$, (**b**) *dy* along $y = x2$, and (**c**) *dz* along $z = x3$.

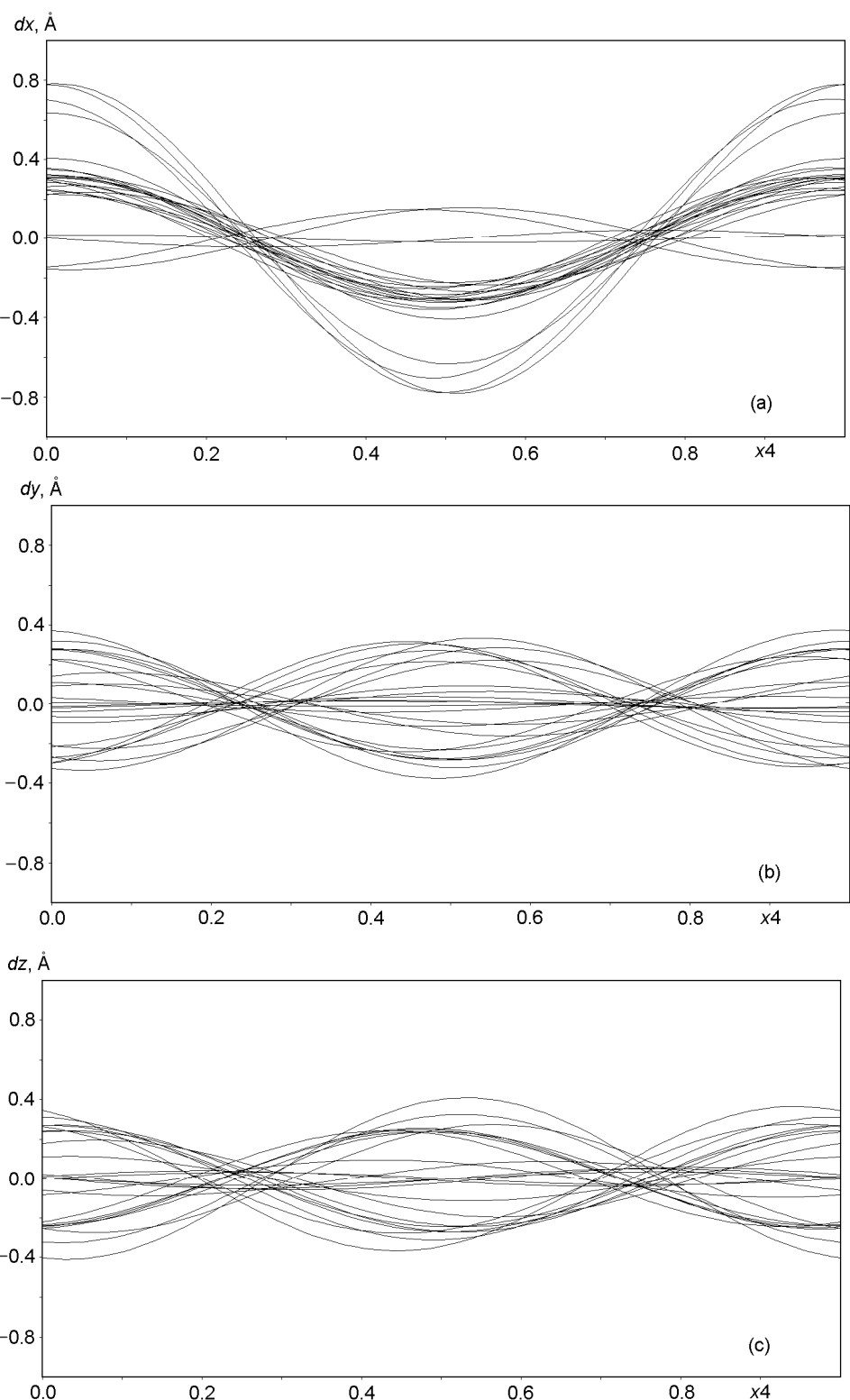

**Figure 7.** Displacements of the atoms O1 through O24 from sites in the basic lattice on a modulation wave period of $0 \leq x4 \leq 1$: (**a**) $dx$ along $x = x1$, (**b**) $dy$ along $y = x2$, and (**c**) $dz$ along $z = x3$.

Harmonic positional modulations of Na and Ca cations are shown in Figure 8. Six of the eight independent cationic sites are occupied by Na, the Ca1 site contains only Ca, and the (Na2, Ca2) site is occupied with two kinds of cations in the Na:Ca = 4:1 ratio. The modulations of Na and Ca differ in their natures from the modulations of the framework atoms. Six Na atoms show positional modulations mainly along the $x1$ and $x3$ coordinates

(Figure 8a,c), whereas atoms at the mixed-occupied (Na2, Ca2) site and Ca atoms at the Ca1 site modulate mainly along the *x2* coordinate (Figure 8b). Modeling of the behavior of the extraframework $SO_4^{2-}$ and $S_3^{\bullet-}$ groups is discussed in the next section.

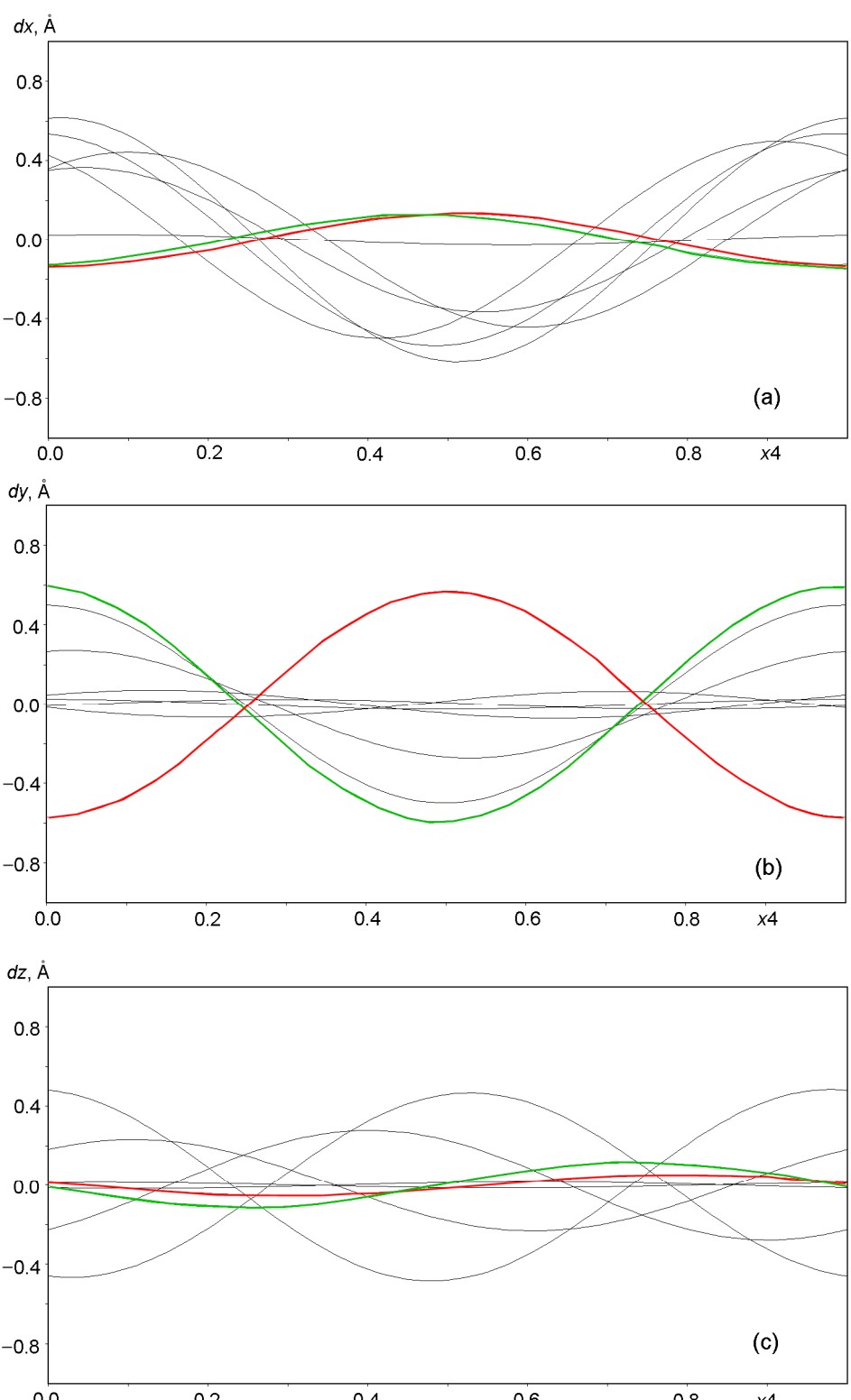

**Figure 8.** Displacements of Na and Ca atoms from positions at nodes in the basic lattice on a modulation wave period of $0 \leq x4 \leq 1$: (**a**) *dx* along $x = x1$, (**b**) *dy* along $y = x2$ and (**c**) *dz* along $z = x3$. The displacements of Ca1 and atoms at the mixed (Na2, Ca2) site are highlighted in red and green, respectively.

A 3D fragment of the modulated structure of the monoclinic LRM is shown in Figure 9. The modulation of $SO_4$ tetrahedra in the row at the level $y \approx 0.5$ looks like a 180-degree rotation around the $c$ axis, when the tetrahedron vertices and its center change their positions abruptly. This can be described in terms of occupancy modulation using special functions called crenels, which are defined on the $x4$ axis and take two values, one and zero [28]. The crenel is equal to one if an atomic position ($x1$, $x2$, $x3$, $x4$) is 100% occupied and equal to zero if the position is empty. The behavior of each atom of the $SO_4$ tetrahedron, which moves abruptly from one position to another, is described by two crenels. The first is equal to one on the interval of length ($s < 1$), centered on $x40$, and the second is equal to one on the adjacent interval ($1 - s$), centered on $x40 + 0.5$. The parameters $x40$ and $s$ are refined in the structural model.

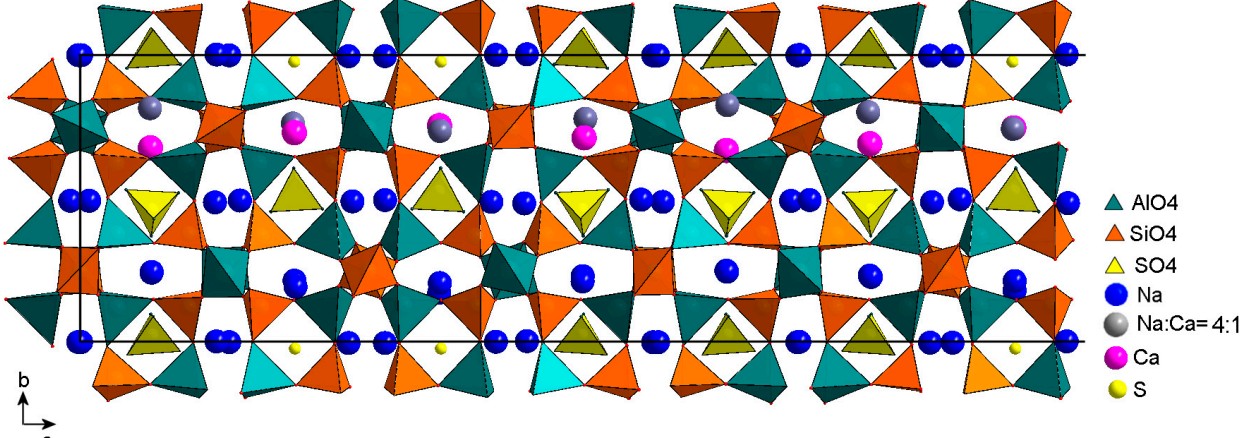

**Figure 9.** Fragment of the monoclinic LRM structure in projection on the plane bc. The length of the fragment along the $c$ axis is $\approx 3/2$ periods of the modulation wave.

In the horizontal row at the level $y \approx 0.5$, differently oriented $SO_4$ tetrahedra centered via S1 and S2 exist on adjacent half-intervals ($s = 0.5$), forming a sequence of fourteen tetrahedra (DUUDDDUUDDDUUD) and covering three periods of the modulation wave. The letters D and U denote tetrahedra with their vertices oriented downwards (D) and upwards (U). The first seven of them are shown in Figure 9. The sequence remains unchanged in the case of commensurate modulation with a wave vector **q** = 3/7, but it is sometimes broken if the length of the wave vector is slightly different from 3/7. Modulations of $SO_4$ tetrahedra are most likely not limited to rotations around the $c$ axis, however, attempts to take into account independent positional modulations for each vertex in the model have resulted in distortions of the tetrahedra without improving the $R$ factor. In the final model, positional modulations are not taken into account, and admissible shapes and sizes of $SO_4$ tetrahedra are provided by setting restraints for the S–O and O–O distances.

Two sulfur positions, S3 and S4, are also localized at the centers of the sodalite cages at the level $y \approx 0$, but the tetrahedron centered with S4 is confirmed only on one half-period of the modulation wave. In the second half-period, only the existence of the S3 center was confirmed. According to IR spectroscopy data for a similar sample (Sample 3), a significant part of sulfide sulfur occurs as the noncyclic $S_3^{\bullet-}$ radical anion. Most likely, the S3 atom is the S–S–S corner vertex, and the positions of the terminal atoms are difficult to localize due to the disordering of the radical anion around S3. According to the chemical analysis data, there is 0.99 sulfide sulfur per formula unit, of which 0.33 is sulfur in the S3 position. We provided the required amount by placing S3 on one-third of the modulation wave period, i.e., two-thirds of the remaining half-period. The probable presence of the $S_3^{\bullet-}$ radical ion in the cavity is indicated in Figure 9 by the central sulfur atom.

## 5. Discussion: Relationships between the Composition and Structures of LRMs

Data on the chemical composition, unit cell parameters, and structural symmetry of three anisotropic and two cubic LRMs are presented in Table 4. Fragments of two one-dimensionally modulated structures are shown in Figures 10 and 11 in projection along the *a* axis. It is convenient to compare these structures with that of the monoclinic LRM (Figure 9) if we choose the parameter $c \approx a_{cub}\sqrt{2}$ of the basic unit cell of monoclinic LRMs as a common measure, as was done in the introduction when discussing the diffraction patterns in Figure 2.

**Table 4.** Data on selected sodalite-group minerals.

| Mineral | Unit-Cell Values $a. b\ c$ (Å) $\alpha, \beta, \gamma$ (°) | Symmetry Group | Modulation Vector | References |
|---|---|---|---|---|
| Lazurite (FK [1], Sample 1) | $a_{cub}$ = 9.087(3) | *P*23 (by analogy with Sample 5) | q ~ 0.30c* (orthorhombic setting) | [4,6] |
| Orthorhombic LRM (vladinirivanovite, Sample 2) $a \sim a_{cub}$, $b \sim a_{cub}\sqrt{2}$ $c \sim 3a_{cub}\sqrt{2}$ | $a$ = 9.057 $b$ = 12.843 $c$ = 38.513 | *Pnaa* | q = 0.33c* | [8,16,17] |
| Monoclinic LRM (similar to Sample 3) $a \sim a_{cub}$, $b \sim c \sim a_{cub}\sqrt{2}$ | $a$ = 9.0692(1) $b$ = 12.8682(1) $c$ = 12.8725(1) $\gamma$ = 90.186(1) | *P*11*a*(00δ)0 | q ~ 0.43c* | [18,19], this work |
| Triclinic LRM (slyudyankaite, Sample 4) $a \sim a_{cub}$, $b \sim a_{cub}\sqrt{2}$ $c \sim 2a_{cub}\sqrt{2}$ | $a$ = 9.0523(4) $b$ = 12.8806(6) $c$ = 25.681(1) $\alpha$ = 89.988(2) $\beta$ = 90.052(1) $\gamma$ = 90.221(1) (*T* = 170 K) | *P*1 | q = 0.5c* | [10,15] |
| Cubic LRM (SO3-bearing haüyne, MD [1], Sample 5) | $a_{cub}$ = 9.077(1) | *P*23 | q ~ 0.43c* (orthorhombic setting) | [6,11] |

[1] Designations from [6].

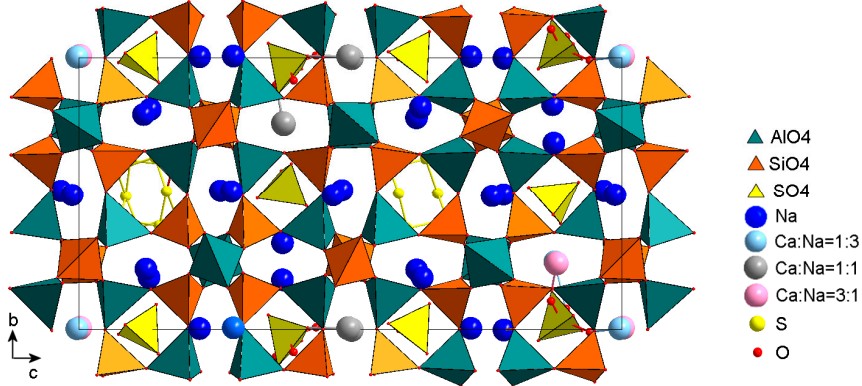

**Figure 10.** The crystal structure of slyudyankaite in projection on the plane *bc*. The S6 molecules and SO4 tetrahedra with different orientations are shown. Some cavities are occupied with tetrahedra in two orientations. Vertices of alternative tetrahedra are marked with red balls. The picture was drawn using data from [10].

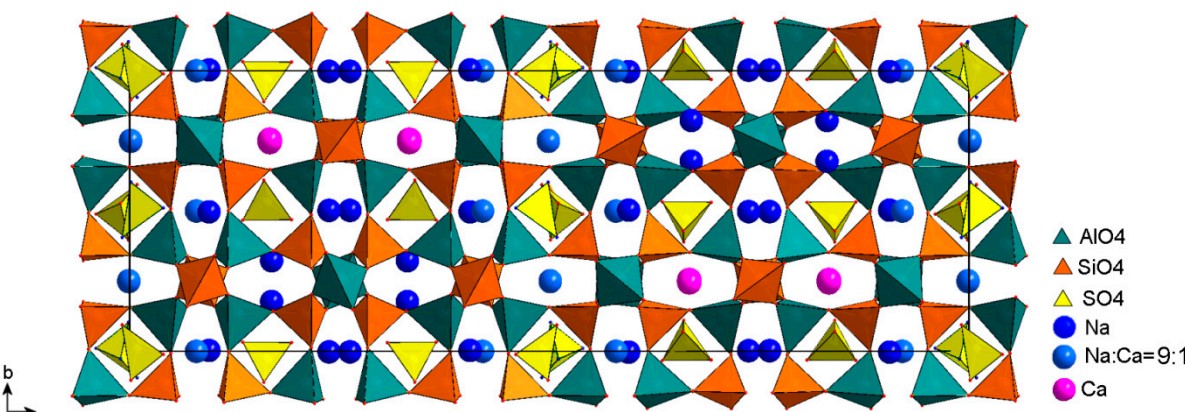

**Figure 11.** General view of the crystal structure of vladimirivanovite. Drawn using data from [16].

### 5.1. The Structure of Triclinic LRMs (slyudyankaite)

Figure 10 shows the crystal structure of slyudyankaite, a triclinic LRM with the commensurate modulation period $2c \approx 2a_{\text{cub}}\sqrt{2}$ [10]. Of the four cages in a horizontal row at the level $y \approx 0.5$, two cages were occupied by $SO_4$ tetrahedra. The other two were partially occupied by cyclic $S_6$ molecules, which are the main species-defining component of slyudyankaite. In the structure of slyudyankaite, there were no positions occupied only by calcium. The relative amounts of the different valence $Na^+$ and $Ca^{2+}$ ions in mixed positions are in good agreement with the amounts of differently oriented $SO_4$ tetrahedra in the corresponding cages. Note that only one angle $\gamma = 90.221(1)°$ noticeably differed from 90°, which may indicate the monoclinic symmetry of the crystal. According to the results of the refinement of the triclinic model [10], for most atoms with coordinates ($x$, $y$, $z$) it is possible to select a pair of the same sort near ($x + 1/2$, $y$, $-z$), but the accuracy of the correspondence is often out of the margin of error. In addition, the monoclinic $P11a$ symmetry is obviously broken by the asymmetric distribution of the $S_6$ molecules in the framework cavities.

### 5.2. The Structure of Orthorhombic LRMs (vladimirivanovite)

The structure of vladimirivanovite (Figure 11) was first studied in the *Pnaa* symmetry group on a single-crystal sample from Southwestern Pamir [16]. At that time, this mineral did not yet have its own name and was considered to be an orthorhombic variety of lazurite. Later, this orthorhombic LRM was characterized using various methods [8,17], including powder diffraction on samples from the Tultui gem lazurite deposit situated in the southwestern Baikal region. The orthorhombic symmetry served as the basis for identifying these crystals as a new mineral species called vladimirivanovite [8].

It should be noted that the structure model of vladimirivanovite does not contain split positions of Al, Si, and O framework atoms; the framework corresponds well to the *Pnaa* symmetry, but almost all cation sites are split into pairs of subsites separated by 0.7–1.2 Å [16]. In Figure 11, their positions are averaged.

### 5.3. Minerals with One-Dimensional Structure Modulation: Similarities and Differences

Before discussing obvious differences in the nature and distribution of extraframework components over the framework cavities of LRMs, let us pay attention to a less obvious similarity between the three structures. Among the Na and Ca cations shown in Figures 9–11 in the gaps of the six-membered rings of tetrahedra, in each figure, it is easy to detect pairs located in the gap of one ring, with visible differences in the $y$-coordinate. Centers of four-membered rings above and below are always occupied by $SO_4$ tetrahedra, each of which is turned so that one of its vertices is directed away from the nearest cation. In other cases, when the positions of cations are superimposed on each other, i.e., do not differ or slightly differ in the $y$-coordinate, both $SO_4$ tetrahedra are turned with their vertices

towards the cations. Moreover, the shapes of six-membered rings with pairs of cations diverging along the *y*-coordinate differ from those with superimposed pairs of cations.

It is not easy to localize the coordinates of $SO_4$ tetrahedra vertices, especially in the structures of monoclinic LRMs with incommensurate modulations. Based on the results of the comparative analysis, one can state, first, that the positions of sulfate sulfur tetrahedra in the framework cavities of each of the three structures are localized reliably. Second, these positions in each of the three structures are similar in orientation in a similar anionic environment.

If the modulations of both the framework atoms and the atoms in the framework cavities basically obey the rules common for the three structures, then the reason for the differences in the modulation periods should be sought elsewhere. For example, in the way various forms of sulfur and various kinds of cations are ordered in the cavities of the framework, or in small differences in the chemical composition. Slyudyankaite (triclinic LRM) contains more Na and less Ca than other LRMs, which leads to the formation of voids surrounded only by $Na^+$ ions, and is sufficient in volume to accommodate large $S_6$ molecules (Figure 10). The structure of slyudyankaite is, in a certain sense, easier to analyze because it does not contain split cation sites, and the amounts of differently oriented tetrahedra in a cavity correlate well to the relative amounts of Na and Ca atoms at mixed sites [10].

The layer of cations in the structure of monoclinic LRMs (Figure 9) at the level $y \approx 0.25$ consists only of $Na^+$ cations in the Na1 and Na5 positions, whereas the layer at the level $y \sim 0.75$ consists of $Ca^{2+}$ cations in the Ca1 position and of different cations in the mixed (Na2, Ca2) position, occupied in the proportion Na:Ca = 4:1. Thus, all calcium occurs in one layer parallel to the *c* axis. The charge neutralization in this layer is provided by the layer of $SO_4^{2-}$ anions. The total charge of the one-valent $Na^+$ cations is neutralized by the $SO_4^{2-}$ anions, which occupy half of their layer, and the $S_3^{\bullet-}$ radical anions in the same layer.

The vladimirivanovite unit cell contains twelve sodalite cages in two rows parallel to the *c* axis (Figure 11). In contrast to triclinic slyudyankaite and monoclinic LRMs, the atoms in different *y*-layers of the vladimirivanovite structure are connected through symmetry elements of the orthorhombic group *Pnaa*; therefore, the cavities and their cationic environment in different *y*-layers do not differ in composition. The cages containing the inversion center are populated with $SO_4$ tetrahedra in two orientations that were realized with the same probability. In addition to symmetry, vladimirivanovite is distinguished by the presence of $SO_4$ tetrahedra in all sodalite cages. The cavities with tetrahedra in one orientation and the central S1 atom are 80% occupied by $SO_4$ tetrahedra, but tetrahedra with S2 centers are distributed over other cavities in one of the two orientations connected to the inversion center with a probability of 20%. Sulfide sulfur is also present in all cavities in small amounts, presumably in the $S_3^{\bullet-}$ and $S_2^{\bullet-}$ forms, but its localization is often difficult. It can be assumed that the disordering of different extraframework components over sodalite cages of various shapes contributes to the lengthening of the modulation period.

### 5.4. Cubic LRMs with Complicated Structure Modulation

In the diffraction pattern of the cubic LRM (Sample 5), the average structure of which was studied by Rastsvetaeva et al. [11], strong satellites with fractional indices $h \pm \delta, k \pm \delta, l$), $(h \pm \delta, k, l \pm \delta)$ and $(h, k \pm \delta, l \pm \delta)$, $\delta \approx 0.2154(1) \approx 3/14$ were observed by Bolotina et al. [12] at the ends of wave vectors, oriented in six directions. Satellites with indices $(h \pm 2\delta, k \pm \delta, l \pm \delta)$ can be explained using a superposition of two waves oriented along different directions [110] and [101] in a cubic lattice, whereas some other satellites can only be explained using a superposition of three differently directed waves. The latter served as an argument for modeling this structure in the (3+3)D space [14]. Somewhat earlier [12], the same structure was studied as a twin of three (3+2)D orthorhombic components.

The (3+3)D model for the cubic LRM (Sample 5 in Table 4) contains three wave vectors of equal lengths: $\mathbf{q}_1 = \delta(\mathbf{a}_{cub}^* + \mathbf{b}_{cub}^*)$, $\mathbf{q}_2 = \delta(\mathbf{a}_{cub}^* + \mathbf{c}_{cub}^*)$, and $\mathbf{q}_3 = \delta(\mathbf{b}_{cub}^* + \mathbf{c}_{cub}^*)$;

$\delta = 0.2154(1)$ [14]. In the corresponding orthorhombic setting of the coordinate axes, any of these vectors can be represented as $\mathbf{q} \approx 0.43\mathbf{c}^*$. Modulation of proper lazurite (Sample 1 in Table 4) occurs in the same main directions with wave vectors of shorter lengths: $\mathbf{q}_1 = \delta(\mathbf{a}_{cub}^* + \mathbf{b}_{cub}^*)$, $\mathbf{q}_2 = \delta(\mathbf{a}_{cub}^* + \mathbf{c}_{cub}^*)$, and $\mathbf{q}_3 = \delta(\mathbf{b}_{cub}^* + \mathbf{c}_{cub}^*)$; $\delta = 0.1479$ [4] or $\mathbf{q} \approx 0.30\mathbf{c}^*$ in the orthorhombic setting. In [12], with reference to [29], the sequence of changes in the diffraction pattern of one of the samples of cubic LRMs during long-term annealing at 550 °C is described. During the first three days of the experiment, satellite reflections with the incommensurability parameter $\delta = 0.217$ coexisted with satellites that had the same indices but a different $\delta$ parameter of 0.147. During the first ten days, the intensities of the former satellite reflections gradually decreased, whereas the intensities of the latter satellites increased. Then, the intensities of satellites of the second type started decreasing until all of the satellites had completely disappeared after two months of annealing. Apparently, for some time, domains characterized by different periods of incommensurate structural modulation coexisted in the sample. Similar processes were observed for other samples of LRMs and were explained using complex mutual transformations of sulfate and sulfide groups [25,27].

*5.5. General Remarks*

Commensurate and incommensurate structure modulations of LRMs supposedly arise during the recrystallization of their early generations [9,10,25,30]. In our opinion, the modulations are due to violations of stoichiometry and (in the case of incommensurate modulations) translational periodicity as a result of substitutions of anions with S-bearing species in which sulfur has a reduced oxidation degree ($SO_3^{2-}$, $S^{2-}$, $S_2^{\bullet-}$, $S_3^{\bullet-}$, COS, $S_4$, $S_6$, $S_2^{\bullet-}$, $S_3^{\bullet-}$, $HS^-$, etc.). Such substitutions result in local distortions of the framework, as well as variable contents and irregular distributions of the extraframework cations $Na^+$ and $Ca^{2+}$ in the structures. Relative contents of different S-bearing species depend on the temperature of recrystallization and the redox conditions, as well as the charge-balance requirements.

## 6. Conclusions

The structural study of monoclinic LRMs completed in this work made it possible to observe from a general standpoint lazurite and LRMs, which were previously known as varieties of lazurite, differing in symmetry and type of structural modulation. Two LRMs, previously known as "triclinic lazurite" and "orthorhombic lazurite", received the status of separate mineral species with the names slyudyankaite and vladimirivanovite due to specific features in their structures or composition. Slyudyankaite contains unique $S_6$ six-membered rings in the framework cavities, and vladimirivanovite is distinguished by the orthorhombic symmetry of its structure. In sodalite cages of these minerals, polysulfide groups ($S_6$ and $S_3^{\bullet-}$, respectively) alternate with sulfate anions. The structures of slyudyankaite, monoclinic LRMs, and vladimirivanovite modulate according to similar rules but with different periods of the modulation wave. The structure of slyudyankaite, the most ordered of the three, modulates with a period of $2c$ (wave vector $\mathbf{q} = 0.5\mathbf{c}^*$), whereas the least-ordered structure, vladimirivanovite, modulates with a period of $3c$ ($\mathbf{q} = 0.33\mathbf{c}^*$). The monoclinic LRMs are located between them, both in terms of the degree of ordering of cations and S-containing components and in terms of the length of the modulation period ($\approx 2.33c$; $\mathbf{q} \approx 0.43\mathbf{c}^*$), which is most likely incommensurate with the period of the basic lattice. Note that the most-ordered structure is the least symmetric (triclinic), and the increase in symmetry to monoclinic and then to orthorhombic is accompanied by the disordering of atoms in the cavities of the framework.

In cubic LRMs, the modulation vectors shorten after annealing from $0.42c^*-0.44c^*$ to $0.30c^*-0.33c^*$ (in an orthorhombic setting). It can be assumed that annealing results in the disordering of sulfur forms in the cavities of the framework, similar to their disordering in orthorhombic LRMs. Prolonged annealing leads to the complete disordering of S-bearing extraframework components and disappearance of modulations.

**Author Contributions:** Conceptualization, N.B.B.; methodology, N.B.B., N.V.C., A.N.S. and M.F.V.; validation, A.N.S. and N.V.C.; investigation, N.B.B., N.V.C., A.N.S. and M.F.V.; writing—original draft preparation, N.B.B.; writing—review and editing, A.N.S. and N.V.C.; visualization, N.B.B. and N.V.C. All authors have read and agreed to the published version of the manuscript.

**Funding:** The crystal structure refinement, Raman spectroscopic studies, and crystal chemical analysis by N.B.B., N.V.C., and M.F.V. were supported by the Russian Science Foundation (grant No. 22-17-00006). Data on infrared spectra and chemical compositions were obtained in accordance with the state task (state registration number: AAAA-A19-119092390076-7).

**Conflicts of Interest:** The authors declare no conflict of interest.

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
