# Peer review of "Structure Modulations and Symmetry of Lazurite-Related Sodalite-Group Minerals"

_crystals, doi:10.3390/cryst13050768_

Round 1
Reviewer 1 Report
. The work conducted by Bolotina et al provides a detailed structural analysis of sodalite-group minerals. Their study is conducted using IR, Raman, and X-ray techniques, with each peak being assigned to a corresponding structure. This analysis of the newly discovered mineral is valuable, and I recommend accepting the paper after minor revisions are made.
1. It is important to compare the compounds with an optically polarizable microscope to determine the opaqueness, crystallinity, and color.
2. Lengthy sentences should be avoided.
3. The peak positions on IR and Raman should be labeled in Figures 4 and 5, and assigning peak modes over the figures could enhance readers' understanding.
4. There should be a space after the '+' or ':' symbols on lines 248 and 262.
5. The figures labelled 6-8 do not correspond to their assigned letters (c, b, c). It is recommended to modify this labelling. One suggestion to improve the presentation of the figures would be to merge figures a, b, c, etc. into a single figure and arrange them in rows and columns within a single box. This would enhance the clarity and organization of the figures.
The English are not bad, but I see too many long sentences. Those sentences should be split to make them smaller.
Author Response
Reviewer 1
Comments and Suggestions for Authors
The work conducted by Bolotina et al provides a detailed structural analysis of sodalite-group minerals. Their study is conducted using IR, Raman, and X-ray techniques, with each peak being assigned to a corresponding structure. This analysis of the newly discovered mineral is valuable, and I recommend accepting the paper after minor revisions are made.
Our response
We are grateful to Reviewer_1 for a careful reading of the manuscript and a positive assessment of our work.
- It is important to compare the compounds with an optically polarizable microscope to determine the opaqueness, crystallinity, and color.
Our response
These data were published earlier (see, for instance, ref. [6] and references therein) and are not directly related to the topic of this work.
- Lengthy sentences should be avoided.
Our response
We have shortened lengths by breaking up long sentences and simplified complex sentences. Changes are highlighted in red.
- The peak positions on IR and Raman should be labeled in Figures 4 and 5, and assigning peak modes over the figures could enhance readers' understanding.
Our response
The exact wavenumber values corresponding to the positions of the peaks in Figures 4 and 5 are shown in Tables 1 and 2, respectively. We think the figures will be overloaded and unlikely to improve understanding if all these peaks are labeled.
- There should be a space after the '+' or ':' symbols on lines 248 and 262.
Our response
Unfortunately, we were not able to number the lines of the template intended for revision in order to see lines 248 and 262. Let us hope the editor notices and corrects this oversight.
- The figures labelled 6-8 do not correspond to their assigned letters (c, b, c). It is recommended to modify this labelling. One suggestion to improve the presentation of the figures would be to merge figures a, b, c, etc. into a single figure and arrange them in rows and columns within a single box. This would enhance the clarity and organization of the figures.
Our response
The following sentence ‘The displacements of Al and Si atoms along the other two coordinates are not so strong in amplitude (Figs. 6b and 6c) but are consistent along the x3 coordinate.’ is replaced with ‘Other displacements of the Al and Si atoms are not so large in amplitude, but they coincide in the x3 coordinate (Fig. 6c). We agree with Referee that it would be good to combine the nine graphs by placing figures 6, 7 and 8 in three columns. When preparing the manuscript, we tried to implement this idea, but due to the large margins adopted in the journal, the graphics were too small, with details that were difficult to distinguish.
Reviewer 2 Report
In this work, the structure of incommesurately modulated monclinic lazurite-related minerals is re-examined based on the superstructure of slyudyankaite formerly known as triclinic lazurite. Similarities and differences between three one-dimensionally modulated lazurite-related minerals and cubic lazurite-related minerals structures modulated in several directions are discussed. Assumptions are made on how the symmetry of the structure and the composition of the crystal can affect the period of structural modulation. The structural study of monoclinic lazurite-related minerals completed in this work made it possible to look from a general standpoint at lazurite and lazurite-related minerals, previously known as varieties of lazurite, differing in symmetry and type of structural modulation. Two lazurite-related minerals, previously known as “triclinic lazurite” and “orthorhombic lazurite”, got status of separate mineral species with the names slyudyankaite and vladimirivanovite due to specific features of their structures or composition. The paper is well motivated, clearly written, original and timely. It is likely to attract the interest of readers of the Crystal journal. The results support excellently the conclusions drawn by the Authors. Hence, I recommend publishing the paper as it stands
Author Response
Reviewer_2
Suggestions for Authors
In this work, the structure of incommesurately modulated monoclinic lazurite-related minerals is re-examined based on the superstructure of slyudyankaite formerly known as triclinic lazurite. Similarities and differences between three one-dimensionally modulated lazurite-related minerals and cubic lazurite-related minerals structures modulated in several directions are discussed. Assumptions are made on how the symmetry of the structure and the composition of the crystal can affect the period of structural modulation. The structural study of monoclinic lazurite-related minerals completed in this work made it possible to look from a general standpoint at lazurite and lazurite-related minerals, previously known as varieties of lazurite, differing in symmetry and type of structural modulation.
Two lazurite-related minerals, previously known as “triclinic lazurite” and “orthorhombic lazurite”, got status of separate mineral species with the names slyudyankaite and vladimirivanovite due to specific features of their structures or composition. The paper is well motivated, clearly written, original and timely. It is likely to attract the interest of readers of the Crystal journal. The results support excellently the conclusions drawn by the Authors. Hence, I recommend publishing the paper as it stands.
Our response
We are grateful to Reviewer_2 for a careful reading of the manuscript and a high appreciation of our work.
Reviewer 3 Report
In their contribution "Structure Modulations and Symmetry of Lazurite-related Sodalite-group Minerals", Bolotina et al. report on the structural analysis of a mineral within the aforementioned series. In addition to the structure determinations, the obtained structure model is compared to those structures which were previously reported in the literature for compounds belonging to this particular sort of minerals. Although the contents presented in this remarkable contribution appear to agree with the scope of Crystals, yet, there are certain issues which should be revised prior to a publication of this work:
- It is stated that X-ray diffraction data were collected for sample 3; however, the authors did not provide any details regarding the integration and absorption correction of the collected data. Because this information is quite relevant to evaluate the quality of the structure solution and refinement, I strongly recommend to include this information in the manuscript. Furthermore, the authors did not provide any information how the structure model was refined. For instance, did the authors use the Superflip code? Please provide this relevant information.
- It is mentioned that chemical analyses of the samples were completed based on EDS and WDS; however, I could not find any spectra in the manuscript. Please provide this relevant data in order to confirm the proposed compositions.
- To distinguish the respective locations of silicon and aluminum atoms within a given structure model, it can also be helpful to collect NMR data and to carry out first-principles-based simulations. Do the authors plan to use such means in order to confirm the proposed structure models?
- With regard to such minerals, the Lowenstein rule is typically taken into consideration. To what extent do the structure models of the reported minerals follow this straightforward idea?
Author Response
Reviewer_3
Comments and Suggestions for Authors
In their contribution "Structure Modulations and Symmetry of Lazurite-related Sodalite-group Minerals", Bolotina et al. report on the structural analysis of a mineral within the aforementioned series. In addition to the structure determinations, the obtained structure model is compared to those structures which were previously reported in the literature for compounds belonging to this particular sort of minerals. Although the contents presented in this remarkable contribution appear to agree with the scope of Crystals, yet, there are certain issues which should be revised prior to a publication of this work:
Our response
We are grateful to Reviewer_3 for a careful reading of the manuscript and a positive assessment of our work.
- It is stated that X-ray diffraction data were collected for sample 3; however, the authors did not provide any details regarding the integration and absorption correction of the collected data. Because this information is quite relevant to evaluate the quality of the structure solution and refinement, I strongly recommend to include this information in the manuscript. Furthermore, the authors did not provide any information how the structure model was refined. For instance, did the authors use the Superflip code? Please provide this relevant information.
Our responce
We did not use Superflip to solve the structure. As stated in the ‘Samples and experimental methods’ section, the average structure of monoclinic LRM was first solved in 2006, and a year later, in 2007, the first data on modulation of the framework were published. But only now the structure has been studied in full, with the refinement of the modulation parameters of all extra-framework atoms.
The data on monoclinic LRM were collected in 2006 in a way typical of a CAD-4 diffractometer with a point detector. Main reflections and first-order satellites were brought one after another on the Ewald sphere. The intensities were measured in the angular interval [omega – delta(omega), omega + delta(omega)]. The deviation from the standard technique consisted in an additional rotation of each reflection by a pre-calculated angle psi around the normal H to the reflecting plane in order to exclude the superposition of close satellites. To take absorption into account, the shape of the sample was approximated by a sphere 0.1 mm in diameter. This information has been added in blue to the text of the paper.
- It is mentioned that chemical analyses of the samples were completed based on EDS and WDS; however, I could not find any spectra in the manuscript. Please provide this relevant data in order to confirm the proposed compositions.
Our response
These data were published earlier (see refs [5-7] in the paper) and are not directly related to the topic of this work.
- To distinguish the respective locations of silicon and aluminum atoms within a given structure model, it can also be helpful to collect NMR data and to carry out first-principles-based simulations. Do the authors plan to use such means in order to confirm the proposed structure models?
Our response
We do not plan any special studies to deal with the distribution of silicon and aluminum atoms within the given structure model. The structure of monoclinic lazurite studied in this work does not contain mixed (Al, Si) positions. They are also not present in the structure of triclinic LRM (slyudyankaite) and orthorhombic LRM (vladimirivanovite), as previously established by structural analysis. The ratio Al:Si is close to 1:1 in three crystals. More precisely, the chemical composition of the framework is Al(6-x)Si(6+x)O24 (Z=1), with the highest value x=0.16 in vladimirivanovite.
- With regard to such minerals, the Lowenstein rule is typically taken into consideration. To what extent do the structure models of the reported minerals follow this straightforward idea?
Our response
As follows from the above, Lowenshtein rule is not violated. However, this rule can be significantly broken in other natural and synthetic microporous materials related to cancrinite and sodalite, see ref. [7] in this work.